# A Case Report of Commercial Production from High Fecundity Livestock in a Pastoral Environment

**DOI:** 10.3390/ani15111583

**Published:** 2025-05-29

**Authors:** Leo James Cummins

**Affiliations:** Independent Researcher, 36 Skene St., Hamilton, VIC 3300, Australia; leocummins46@gmail.com

**Keywords:** commercial lamb and beef production, *FecB* sheep, USMARC Twinner cattle

## Abstract

This report describes the commercial use in a pastoral situation over a number of years, of including sheep and cattle genotypes with higher-than-normal ovulation rates. In both species, the use of these genotypes has increased the number of offspring weaned by around 30%.

## 1. Introduction

### 1.1. Grazing in South-Western Victoria

Compared to most other parts of Australia, South West Victoria has a reliable climate, relatively fertile soils, and quite good pastures, and thus might be considered suitable for higher fecundity grazing livestock. Average annual rainfall varies from 550 to 900 mm, and the growing season is about 7–9 months with a spring peak. Around 67% of the area is devoted to livestock grazing enterprises. The pastures typically have a moderate sub-clover/perennial pasture grass base [1]. Pasture production is highly variable and dependent on rainfall and pasture management, but on typical farms, a long-term annual average of 7–10 tonnes of dry matter can be expected. The seasonal nature of pasture production means that the timing of reproduction and turn-off for sale animals needs to be carefully considered. Continued farm production needs to consider the suitability of the final consumer product and the social license to produce it, as well as the economics of production, all of which are liable to change over time. An Agriculture Victoria livestock farm monitor project [2] involving up to 60 operating farms in the South West region and extending over 20 years, indicates only moderate profitability with an average gross margin of $620/hectare and a return to equity of 3.1%, but with very large variations between farms and between years. In these data, wool (Merino-based) flocks had an average lambing percentage of 79%, prime lamb-producing flocks (mostly cross-bred ewes) had an average lambing percentage of 111%, and for beef cattle, the calving percentage was 89%. Ewe fertility did improve slightly over this period, probably due in part to the application of genetic, nutrition, and other management practices, but cow fertility barely changed.

Profitable farming requires optimization of all facets of production, returns, and costs. Significant changes in any one aspect of production, such as fecundity, will almost certainly require some modifications to other aspects, which may be difficult to apply on commercial farms. This report gives details of a commercial, genetically high fecundity livestock enterprise in South-Western Victoria. Detailed farm modeling in this environment has shown that higher fecundity should reduce the methane emissions intensity and increase farm output in both prime lamb [3] and beef enterprises [4]. This is largely due to changes in flock and herd structure, i.e., fewer breeders and more offspring with an overall total increased output.

### 1.2. Sheep

The *FecB* gene was originally recognized in the Booroola Merino in Australia. This gene arrived in Australia in the “Bengal Sheep” (presumably Garoles, a small-stature highly fecund group) with the first European settlers in the 1790s [5,6]. Merinos and other breeds were soon imported, and the Australian wool industry developed, and the industry expanded rapidly until 1895 when a major drought halved the sheep population. After this, the wool industry again expanded to reach a peak population of 180 million in 1970 [7]. The merino flock in Australia is not monovular, but only 15–30% twin ovulations would be typical in many merino flocks. In typical prime lamb flocks, usually crossbred, 40–60% twin ovulations might be expected. This is dependent on nutrition, day length, and genetics. A long-term lamb marking percentage of around 80% for wool flocks in many of the better areas of Australia would be typical. Over this time, the Australian flock was selected for wool production, but given the flock expansions that occurred, reproduction must have been a consideration. However, the high fecundity of the original sheep seemed to disappear within the industry. An individual farming family, the Sears, established a multiple birth flock on their properties “Mynora-Booroola” in about the 1930s and by the 1960s had a small flock with a 200% lambing rate, which seemingly originated from two highly fecund ewes [8]. A sample of this flock formed the basis of the CSIRO (Commonwealth Scientific and Industrial Research Organisation, Australia) investigation of the Booroola Merino [6,9].

It was identified as a major gene primarily affecting ovulation rate [6,9], and it is a point mutation in the BMP-1B receptor on chromosome 6. In the Booroola merino, the effect of the gene on ovulation rate was additive, with one copy of the gene resulting in a mean ovulation rate of around three, and two copies around four–six. For litter size, one copy improved lambs born by 0.9 and two copies by a further 0.4, but the higher litter sizes, without special husbandry, often had increased lamb mortality. On this basis, the ideal commercial ewe would probably be heterozygous.

### 1.3. Cattle

Twinning in beef cattle is unusual, and the maximum natural fertility is considered to be one calf per cow per year. In many environments, this is very difficult to achieve. Their low incidence means that twinning cows are not identified and managed, and any problems with twins are emphasized by producers and their advisors. The fact that the cost of keeping a cow to produce one calf is a very expensive nutritional exercise with a high methane output is overlooked. Since the 1940s, efforts have been made to increase the twinning rate by hormonal manipulation, embryo transfer, and genetic selection. So far, these have been expensive, somewhat unreliable, and operationally difficult at the farm level, and hence unlikely to be profitable; however, this technology is changing.

The most successful genetic program in beef twinning has been the multibreed/composite selection line developed by the United States Meat Animal Research Center (USMARC) in Nebraska [10,11]. The twinning rate improved from 9% in 1984 to greater than 50% in 2000, a linear rate of improvement of 3% per year. This was a long-term selection program, and the genetic basis of this response is likely to be multigenic and very different for the Booroola sheep.

## 2. Materials and Methods: Sheep

The development of a commercial DNA test for the *FecB* gene allowed the development of a management system for a high fecundity commercial prime lamb producing flock in South-Western Victoria.

The *FecB* gene was introduced to our flock in 1982 through 3 Booroola merino rams used over a line of composite ewes, which originally had a Corriedale, Border Leicester, Coopworth background. These ewes were chosen to improve lamb survival and be more suitable for prime lamb production. Until the early 2000s, the farm enterprise was primarily wool production with a small separate prime lamb enterprise. Initially, the *FecB* gene was maintained by retaining young ewes that had 3 ovulations. The sires used for this flock were purchased (non-booroola) composite rams. When the DNA test became available at the *Genomnz ^TM^* AgResearch DNA genotyping laboratory (at Mosgiel in New Zealand) during the early 2000s, the whole flock gradually moved to become a composite prime lamb flock. The flock then became a semi-closed flock, breeding its own rams and replacement ewes. The rams used were bred in a small nucleus sub-flock, and they were DNA sampled for the *FecB* gene. In this ram breeding sub-flock, approximately 20% of the sires were sourced from other composite high-performing industry flocks (i.e., non-Booroola) to limit inbreeding to some extent, and to introduce genetics to improve worm resistance and carcass value. The rams used in the commercial flock were chosen to be 50% non-*FecB* carriers and 50% heterozygotes. The commercial ewe flock was not assessed for *FecB.* status, but was clearly segregating for the *FecB* allele. This compromise was chosen to simplify the self-contained breeding structure on the farm. The ewes were mated in February, typically on poor pastures with some supplementary feeding. Autumn rains generally improved pasture status by mid-pregnancy when the potential litter size was assessed by ultrasound (this scanning included litter sizes of three or more as a single category). Lambs were sold at weaning in local commercial markets in December, where they generally matched those of many other producers’ lambs from normal, less fecund flocks. Management of these ewes was dependent on fetal counting using ultrasound pregnancy testing to have groups of ewes with singles allocated to the poorer paddocks (based on pasture quality, quantity, and shelter), twins on better paddocks, and those scanned with triplets or more on the best paddocks. Lambing was not closely supervised.

## 3. Results: Sheep

The results of the pregnancy scanning in the commercial flock are shown in Table 1.

For these 5 years, a total of 7152 ewes were scanned, and the overall average litter size was 189%, with around one quarter of the ewes in this flock scanned with three or more lambs.

The lambs were marked (tailed, castrated, vaccinated, and tagged) at 3 to 7 weeks of age, see Table 2.

The perinatal ewe mortality is the difference between the number of ewes allocated to paddocks after scanning and those present at lamb marking time. The overall ewe mortality was 6.4%, which is at the higher end of the typical industry range of 3% to 10%. An industry project [12] where 5.9% of ewes mated were identified as carrying triplets gave survey figures from 105 flocks indicating that the mortality of single-bearing ewes averaged 1.6% (10–90% range 0.5–3%), twin bearers 3.3% (10 to 90% range 1.2–5%), and triplets 6.4% (10–90% range 1.8–14.5%). Clearly, the high litter sizes are a problem. The category of 3+ in this flock is likely to include both heterozygous and homozygous *FebB* ewes. The lamb survival figures for our flock were 97% for the singles, 81% for the twins, and 59% for the triplets. These are within the specific industry survey [12] figures for singles of 92% (10–90% range 86.2–96.8%), for twins of 80% (10–90% range 71–88.6%), and triplets 59% (10–90% range 45.3–70.3%). Again, triplet survival is a problem. These records collected on this commercial farm indicate that the *FecB* gene segregating in a prime lamb-producing flock will increase the likely long-term lambing percentage from 111% up to 145%. The mortality figures confirm that litter sizes of three or more in this environment are a problem which needs to be addressed by more targeted management and nutrition. The most important tool is the ability to predict the number of lambs due. This can be performed using ultrasound, and then the ewes with triplets need to be specifically considered for ewe health and fitness, nutrition (and the way to provide supplements if necessary), predators, and shelter, and only a small number of triplet-bearing ewes are placed in each lambing paddock. These recommendations are the outcome of an ongoing industry survey and research work [12]. Intensive lambing systems in other economic and physical environments can handle higher lambing percentages than seen here. The best commercial use of the *FecB* gene in this environment would seem to be as heterozygous prime lamb dams. This requires a structured breeding program with a small nucleus flock producing homozygous *FecB* rams of a suitable genotype, used to mate with suitable non-carrier ewes to produce the commercial prime lamb dam. This concept has been suggested by CSIRO with the development of the Booroola Leicester and Colin Earle with the Multimeat [13], but with only modest industry acceptance in Australia.

## 4. Materials and Methods: Cattle

Embryos and semen from this line were imported to Australia in several batches, starting in 2004. The embryos were from a Canadian herd which had maintained a line of cattle derived from the USMARC Twinners, while the semen was from 6 bulls purchased directly from USMARC. These were incorporated into a small commercial beef herd based on Angus and Shorthorn cattle. Over time, this herd was upgraded towards the USMARC Twinners line. The use of Embryo Transfer and Artificial Insemination in a small but variable portion in most years continued up until 2018, but since then, these outside options have become limited. Maintaining and further developing this line of cattle, which is very different from normal cattle, presumably mainly due to multigenic control of ovulation rate, poses many similar issues to those faced by the rare breeds. However, the herd is still operating, and from the results included in this paper, I estimate that the cattle are at least 50% or more USMARC Twinner blood.

Our herd calves in Autumn, the calves are weaned in December, and the weaned cattle are sold to the industry in January. These weaners typically go through a further grow-out phase before slaughter. Calving is managed in a system with twice daily inspection to provide assistance at birth and for the first couple of days, as required, to ensure bonding with both calves. Then, the cows with their calves are removed from the calving paddock and managed as a normal herd. Supplementary feed with hay is usual in autumn. At weaning, all heifers are checked for freemartinism. This involves measuring vaginal length with a suitable probe. At this age freemartins have a vaginal length of around 10–12 cm and normal heifers 25–35 cm. (This simple measurement confirms the observation that virtually all the freemartins have been recorded in a male + female birth type). The steer calves and surplus females go into regional weaner sales, where they seem to fit within the range of weights for the calves offered by other local producers. The cows are joined in June with bulls of the same line. These replacement bulls have been selected from the male calves on the basis that they were born as twins or were embryo transfer progeny, are polled, and phenotypically acceptable. In practice, these home-bred bulls have sired most of the calves. The cows are pregnancy tested in August by ultrasound to count the number of fetuses present. This record is very important for future calving management. As the farm is quite small, single- and twin-bearing cows are managed together, while on a larger farm, they would be managed separately to ensure that body condition could be better maintained during late pregnancy and early lactation, and extra calving supervision focused on them. The increased short-term supervision is very similar to that required for heifers having their first calf [14].

For the reproductive rates reported for the heifers, there were 47 twins and 129 singles born over the years 2005 to 2014 and exposed for mating. The reproductive rate was analyzed using a General Linear Model with a multinomial distribution and a logic link function with fitted effects of year and birth type.

## 5. Results: Cattle

The reproductive data in Table 3 and Table 4 are reported on a per-cow calving basis. This is because of culling, which can occur between the start of mating and calving for many reasons, as well as removing empty cows. An economic study in our environment indicated that increases in fertility over 75% in normal herds did not improve farm profitability, provided the empty cows are sold [15]; however, this did not consider the possibility of twinning.

The average weaning weight for the calves produced by the cows shown in Table 3 was 311 kg (SD 45) for singles and 267 kg (SD 49) for twins. The twins each weighed 86% of a single, or a cow successfully having twins produced 172% of the weaning weight of a cow with a single.

The results of a more recent survey (from 2021 to 2023) from five commercial herds with USMARC blood in South Eastern Australia and Canada are shown in Table 4. Each farm has its own unique environment and management.

Table 4 and Table 5 show that in terms of calves born per cow calving, USMARC blood cattle had over 30% twin births. These genetics have resulted from both imports and grading-up, and so do not fully represent the final years of the USMARC project, but they do confirm that these cattle are very different from normal beef cows. The overall calf mortality of 13% is quite high, with a 95% survival rate for singles and 81% for twins. The between-year and between-farm variation is also quite high, suggesting there are likely to be important management and environmental factors which could be addressed. From a commercial point of view, the overall calf weaning percentage of 115% for these USMARC blood cows can be compared with the local industry figure of 89%, an improvement of 26% with the opportunity of further improvement if management and nutrition can reduce the calf mortality to the best groups in the data shown.

To examine the effect of birth type on pregnancy rates in 15-month-old heifers, the results from herd 1 were further examined, as shown in Table 5. There were 176 heifers born between 2005 and 2014.

The pregnancy rates and calving rates in this dataset are relatively low, but the time series includes 3 out of 10 years of low rainfall and poorer pasture production, years in which feeding breeders (both ewes and cows) took priority. Nevertheless, it is clear that being born a twin did not reduce the conception rates or calving rates. The weaning weights of these calves, recorded 7 months prior to this joining, were 312 kg for singles and 256 kg for twins (i.e., the twins weighed 82% of a single), and no later weights were taken. Given the similarity of pregnancy rates and the importance of liveweight on conception rates in heifers, it is likely that the twins caught up in weight over this post-weaning period.

## 6. Conclusions

Genetically high fecundity livestock has increased progeny output from the typical district levels of 111% to 145% for prime lamb production and from 89% to 115% for beef cow production. This is around a 30% improvement in output in both species and could change herd structure and reduce methane intensity significantly. The prediction of fetal numbers (by ultrasound) is available in both species. A better understanding and application of the management requirements of animals with higher litter sizes should lead to reduced neonatal mortality and even further improvements in output.

## Figures and Tables

**Table 1 animals-15-01583-t001:** Ultrasound results: scanned litter size (May).

Year	Scanned Litter Size %				Total No Ewes
	0	1	2	3+	
2014	3	28	43	25	1458
2015	6	30	43	21	1551
2016	5	18	45	33	1310
2017	5	21	51	22	1331
2018	7	29	46	18	1502
Average	5	25	46	24	

**Table 2 animals-15-01583-t002:** Lamb marking (August/September).

	Peri- Natal Ewe Mortality%			Lambs Marked per Ewe Present at Marking%			Overall Marking%
Pregnancy Scan	1	2	3+	1	2	3+	
2014	4	3	13	92	168	184	142%
2015	2	4	7	92	158	183	138%
2016	2	2	8	100	162	191	159%
2017	5	7	17	104	155	176	148%
2018	0	7	15	97	163	157	140%
Average	3	5	12	97	161	178	145%

**Table 3 animals-15-01583-t003:** Reproductive performance of USMARC Twinner cattle * in South-Western Victoria.

	2013	2014	2015	2016	2017	Total/Average
No of Cows	56	82	77	73	72	360/72
Calves born/cow calving	1.37	1.24	1.44	1.40	1.40	1.33
Calves weaned/cow calving	1.25	1.11	1.14	0.99	1.24	1.14
Calf mortality	8%	11%	21%	33% **	12%	14%

* This is a mixed-age herd with approximately 25% first-calf heifers. ** The high calf mortality in 2016 may be a result of severe drought conditions from mid-pregnancy until mid-lactation, causing nutritional stress.

**Table 4 animals-15-01583-t004:** Reproductive performance in several USMARC Twinner herds.

Herd	No. Cows Calving	Calves Born	Calf Mortality	Calves Weaned	No. Cows Requiring Calving Assistance
1 *	100	144 (144%)	29 (20%)	115 (115%)	16 (16%)
2	59	73 (124%)	8 (11%)	65 (110%)	1 (2%)
3	177	231 (131%)	35 (15%)	196 (111%)	8 (5%)
4	7	11 (157%)	0	11 (157%)	0
5	200	258 (120%)	18 (7%)	240 (120%)	8 (4%)

* Herd 1 presents further data from the herd shown in Table 3.

**Table 5 animals-15-01583-t005:** Predicted pregnancy and calving rates in first-calf heifers born as a single or as a twin.

Birth Type	Pregnancy Rate			Calving Rate		
	NP *	Single	Twin	NP *	Single	Twin
Single	21%	61%	18%	36%	54%	11%
Twin	17%	60%	23%	27%	58%	15%

* NP = not pregnant.

## Data Availability

The original contributions presented in this study are included in the article. Further inquiries can be directed to the corresponding author.

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
