# Peer review of "A Case Report of Commercial Production from High Fecundity Livestock in a Pastoral Environment"

_animals, 2025, doi:10.3390/ani15111583_

Round 1

Reviewer 1 Report

Comments and Suggestions for Authors

The reproductive performance of any farm animal species has a high impact on efficient production. In mono-ovulatory or less ovulatory animals, it is a crucial question how we can increase their productivity. The Authors introduce their work in the field using the FEC B gene in sheep and the TWINNER cows.

The title is adequate.

The short summary and abstract are clear.

Maybe the structure of a case study or report is not different from that of a research article. It is recommended to restructure the manuscript as follows:

The „background” could be changed for the „introduction”, which is short but clearly indicates all specific parameters in this field, especially the situation in NSW in Australia. However, it could be extended with part of the content in the results section (sheep and cattle historical elements).

Maybe after these in a new section, Material and methods, it would be beneficial to introduce more details about the circumstances where the experiment was made, exactly how the dam line of sheep was created, and the rams used, and the same in cattle. This section should contain all the methods and protocols used in the flock and herd during the investigation period.

line 137 reference in number or words

Results are highlighted in five tables, which are adequate; in Table 5, the number of investigated heifers  (single and twin) is missing.

All together, the work is tremendous and practical, a lifelong experience, which should be shared.

Nowadays, production efficiency, labour cost, methane emission and environmental footprints are always the questions from different parts of society

Author Response

Comments 1 The title is adequate. Response, I agree

Comments 2;  Maybe the structure of a case study or report is not different from that of a research article. It is recommended to restructure the manuscript as follows:

The „background” could be changed for the „introduction”, which is short but clearly indicates all specific parameters in this field, especially the situation in NSW in Australia. However, it could be extended with part of the content in the results section (sheep and cattle historical elements).

Maybe after these in a new section, Material and methods, it would be beneficial to introduce more details about the circumstances where the experiment was made, exactly how the dam line of sheep was created, and the rams used, and the same in cattle. This section should contain all the methods and protocols used in the flock and herd during the investigation period.

Response; I considered that the original version as a case report on a long term commercial enterprise so the narrative approach would be suitable, but both reviewers suggested that a major rearrangement of this nature would improve the article so I have restructured it as suggested. This has required considerable shuffling around The attached changes are highlighted in red and notified in the attached manuscript. I have also taken the opportunity to extend some of the relevant history. (See attached file.)

Comment 3, line 137 reference in number or words

Response; Changed

Comment 4; Results are highlighted in five tables, which are adequate; in Table 5, the number of investigated heifers  (single and twin) is missing.

Response. The number of twins and singles is added in the material and methods and a further description of the method of analysis added.

I thank the referee for the comments at the end

Reviewer 2 Report

Comments and Suggestions for Authors

Dear Author,

The idea of such a case report is by no means negligible and would be extremely useful in daily practice, especially considering that farms focused on meat production are highly dependent on the offspring conceived. Unfortunately, the article resembles more of a narrative rather than an objective and replicable analysis of field situations, which could otherwise be of great benefit to livestock breeders.

Although this is a case report, it would be highly beneficial to structure it into sections such as Materials and Methods (where a description of the farms and the methods through which the gene was introduced should be provided), Results (where the effects of introducing this gene should be clearly and concisely explained, both in terms of prolificacy and economic impact), Discussion, and Conclusions.

In its current form, the article is difficult to follow and is not suitable for publication.

Author Response

Comments 1; The idea of such a case report is by no means negligible and would be extremely useful in daily practice, especially considering that farms focused on meat production are highly dependent on the offspring conceived. Unfortunately, the article resembles more of a narrative rather than an objective and replicable analysis of field situations, which could otherwise be of great benefit to livestock breeders.

Response; To some extent the article is a narrative covering up to 40 years practical farm experience with available high fecundity genetics, so I felt a narrative as a case report was appropriate, but this seems to have been overruled. It is not meant to be a detailed discussion of FecB or twinners - they already have an extensive series of scientific reports available and in fact the animals here responded more or less (within the limits of farm recording) as expected. Deliberately, the report is largely limited to  reproduction. Other farmers going down this path might well choose many variants, however the persistence of this one example may be helpful to both farmers, their advisors and scientists working in this field

Comment 2;  Although this is a case report, it would be highly beneficial to structure it into sections such as Materials and Methods (where a description of the farms and the methods through which the gene was introduced should be provided), Results (where the effects of introducing this gene should be clearly and concisely explained, both in terms of prolificacy and economic impact), Discussion, and Conclusions.

Response; Both reviewers have  recommended that the report be laid out in the usual  manner so I have revised it accordingly. This has required considerable shuffling and the attached file has the changes highlighted in red and  notified. I have also take the opportunity  to slightly extend the history. It would be very hard to provide realistic economic responses directly from this farm experience. An estimate of the economic impact could come from benchmarking with other local farmers (eg see Ref 1) but even that would be difficult or alternatively detailed modelling such as in refs 2 and 3.

Round 2

Reviewer 1 Report

Comments and Suggestions for Authors

The Author answered all the questions and restructured the Manuscript. May be some additional information could improve the used statistical model and statistical program.

Author Response

Comments The Author answered all the questions and restructured the Manuscript. May be some additional information could improve the used statistical model and statistical program.

Response. Thanks. The statistics were performed by a professional agricultural statistician See (acknowledgements)

Please note that in response to reviewer 2,  I have added a bit more background and a bit more sheep history. I would also emphasize that this is a case report on a practical farm operating over many years and the owner has had an interest in high fecundity stock.

Reviewer 2 Report

Comments and Suggestions for Authors

Dear authors,

The article has improved considerably after the first revision and is beginning to take a publishable form, with clear potential to support the ruminant sector in enhancing its practices and achieving higher levels of performance. However, a careful rereading and general correction of spacing and formatting are still needed. A few more specific suggestions are provided below.

Lijne 21 I suggest deleting this section

Lines 22-41 Bibliographic sources need to be added for the first paragraph, especially regarding the aspects related to climate and traditional pasture.

Line 52 Delete sheep, is not necessary

Line 53 a bibliographic source is necessary here

Line 60 The content within the parentheses should be integrated into the main text

Line 80 delete Cattle:

General comment: I suggest that the Materials and Methods section for sheep should be immediately followed by the corresponding section for cattle, and the same structure should be applied to the Results section.

Lines 98–123: A chronological approach to the events described would greatly enhance the reader’s understanding of both the text and the author’s message. It is essential to state clearly that the discussion refers specifically to sheep, with over 30 years of genetic selection. The exact moment when genetic testing was introduced should also be specified.

Line 138: A bibliographic source should be included here to support the statement

Lines 157-158 I suggest you to replace with this the statement “Intensive lambing systems in other economic and physical environments can achieve similar or slightly higher lambing percentages than those observed in this study, though typically only through extensive hormonal interventions and ewe management practices (Comparative data about estrus induction and pregnancy rate on lacaune ewes in non-breeding season after melatonin implants and intravaginal progestogen).

Lines 166-203 In line with the earlier suggestion regarding sheep, a chronological presentation of the genetic interventions introduced in this case is also recommended. Such an approach would improve the study’s reproducibility and strengthen its potential as a reference model within the field.

The conclusion section is well written and, in my opinion, does not require any adjustments

Author Response

Comments 1

Lijne 21 I suggest deleting this section

Lines 22-41 Bibliographic sources need to be added for the first paragraph, especially regarding the aspects related to climate and traditional pasture.

Response I believe this needs to be included since high fecundity at pasture requires a good ebvironment. I have added a reference here

Comments 2 Line 52 Delete sheep, is not necessary & Line 80 delete Cattle: General comment: I suggest that the Materials and Methods section for sheep should be immediately followed by the corresponding section for cattle, and the same structure should be applied to the Results section.

Response I believe that it very appropriate to separate the sheep and cattle sections, the genetic mechanisms are very different

Comment 3 Line 53 a bibliographic source is necessary here

Response Done

Comment 4 Line 60 The content within the parentheses should be integrated into the main text

Response Done

Comment 5 Lines 98–123: A chronological approach to the events described would greatly enhance the reader’s understanding of both the text and the author’s message. It is essential to state clearly that the discussion refers specifically to sheep, with over 30 years of genetic selection. The exact moment when genetic testing was introduced should also be specified.

Response. extra history for the sheep section has been added

Comment Line 138: A bibliographic source should be included here to support the statement

Response This statement should be obvious from the discussion in the sheep introduction

Comment 6 Lines 157-158 I suggest you to replace with this the statement “Intensive lambing systems in other economic and physical environments can achieve similar or slightly higher lambing percentages than those observed in this study, though typically only through extensive hormonal interventions and ewe management practices (Comparative data about estrus induction and pregnancy rate on lacaune ewes in non-breeding season after melatonin implants and intravaginal progestogen).

Response I don't believe this is necessary. For example Finn sheep have been in many systems for years, ovastim and melotonin work and of course the hormonal stimulation work

Comment 7 Lines 166-203 In line with the earlier suggestion regarding sheep, a chronological presentation of the genetic interventions introduced in this case is also recommended. Such an approach would improve the study’s reproducibility and strengthen its potential as a reference model within the field.

Response. This is not a planned scientific study. It is a case report that has evolved over time, by a farmer who had an interest in high fecundity animals. It shows that these genetics can give higher fecundity (in spite of significant industry skepticism). Other farmers and their advisors might use this as a background but undoubtably many practical variations would be considered